# Digital and Economic Determinants of Healthcare in the Crisis-Affected Population in Afghanistan: Access to Mobile Phone and Socioeconomic Barriers

**DOI:** 10.3390/healthcare9050506

**Published:** 2021-04-27

**Authors:** Jin-Won Noh, Yu-Mi Im, Kyoung-Beom Kim, Min Hee Kim, Young Dae Kwon, Jiho Cha

**Affiliations:** 1Division of Health Administration, College of Software and Digital Healthcare Convergence, Yonsei University, Wonju 220710, Korea; jinwon.noh@gmail.com; 2Department of Nursing, Dankook University, Cheonan 31116, Korea; ymim@dankook.ac.kr; 3Department of Health Administration, Dankook University, Cheonan 31116, Korea; aefile01287@korea.ac.kr; 4Department of Physical Therapy, College of Health Science, Eulji University, Gyeonggi 34824, Korea; kmh12@eulji.ac.kr; 5Department of Humanities and Social Medicine, College of Medicine and Catholic Institute for Healthcare Management, The Catholic University of Korea, Seoul 06591, Korea; snukyd1@naver.com; 6Humanitarian and Conflict Response Institute, University of Manchester, Manchester M13 9PL, UK

**Keywords:** afghanistan, complex emergencies, mobile phone, access to healthcare, cash transfer

## Abstract

Despite recent progress in Afghanistan’s health system from the support of international donors and NGOs, protracted conflicts combined with a series of natural disasters have continued to present substantial health risks. Extreme poverty has still aggravated social determinants of health and financial barriers to healthcare. Little is known about the context-specific factors influencing access to healthcare in the crisis-affected population. Using a subset of data from ‘Whole of Afghanistan Assessment (WoAA) 2019’, this study analyzed 31,343 households’ data, which was collected between 17 July and 19 September 2019 throughout all 34 provinces in Afghanistan. The outcome measured was access to care in the healthcare facility, and multivariable binary logistic regression models were used to identify the specific factors associated with access to healthcare. Of 31,343 households exposed to complex emergencies in Afghanistan, 10,057 (32.1%) could not access healthcare facilities when one was needed in last three months. The access to healthcare was significantly associated with displacement status, economic factors such as employment status or total monthly income, and the distance to healthcare facilities. Significant increase in healthcare access was associated with factors related to communication and access to information, such as awareness of humanitarian assistance availability and mobile phone with a SIM card, while disability in cognitive function, such as memory or concentration, was associated with poorer healthcare access. Our findings indicate that the crisis-affected population remains vulnerable in access to healthcare, despite the recent improvements in health sectors. Digital determinants, such as access to mobile phone, need to be addressed along with the healthcare barriers related to poverty and household vulnerabilities. The innovative humanitarian financing system using mobile communication and cash transfer programs would be considerable for the conflict-affected but digitally connected population in Afghanistan.

## 1. Introduction

Despite a fragile economy and an infrastructure devastated by decades of conflicts and instability, Afghanistan’s health system has made significant improvements in recent years [1,2,3]. In the early 2000s, the maternal mortality ratio was recorded as being at the bottom of the international ranking (1600 per 100,000 live births) and the mortality rates of infants and children both ranked as the fourth highest in the world (165 and 257 per 1000 live births). Less than 10% of the population had access to health services within a one-hour walk to a health facility [4]. Since 2003, however, the Ministry of Public Health (MoPH) has strived to develop the healthcare system through the Basic Package of Healthcare Services (BPHS) and Essential Package of Hospital Services (EPHS) by contracting public health services with non-governmental organizations (NGOs) and a semiautonomous agency within the health ministry [1,5]. These packages have involved significant increases in the number of functioning primary health care facilities and the quality of care in publicly financed facilities, which have enabled the increase of health facility deliveries and in antenatal care from a skilled provider [1,6]. Standard clinical and administrative management have been provided to supply basic health care services [2,4,7].

In spite of substantial progress in the war-torn health system, Afghanistan still faces health challenges due to a lack of sustainable health financing and workforce and a high reliance on international donors [8]. In the last decade, economic hardships have been aggravated in many households, impacting on the social determinants of health through complex pathways for those trapped in poverty [9,10]. The epidemiological transition with non-communicable disease has been an additional burden on the rapidly growing population. In a fragile health system without sustainable health financing, health services are often delayed or inaccessible, resulting in preventive deaths or chronic diseases that result in lost income and higher healthcare costs [6]. Given those economic barriers and extreme poverty, cash transfer programs have been considered as innovative intervention models to increase human capital in poor households [11]. In Afghanistan, international actors have operated pilot cash transfer programs, although they are still limited by risks of corruption, diversion, and inflationary effects in the complex operational environment [12,13,14].

In this context, humanitarian concerns have been raised particularly about the complex health consequences of protracted conflicts, coupled with political instability, and a series of natural disasters. The crisis-affected population has been forcibly and continuously displaced, both cross-border or internally, leaving more than 7 million people at health risk without adequate access to primary health services [3]. In 2019, an average 36,181 persons were displaced each month by conflicts, following the displacement of more than 200,000 due to drought in 2018. There were 430,000 returnees from Iran after its economy was substantially slowed down under sanction [15]. The crisis-affected population is at higher risk of physical and mental illness, including trauma and disabilities, requiring an uninterrupted provision of essential care [2,3,16]. The death toll was the highest in 2018, with almost double the number of civilian deaths and injuries compared with 2009 [17]. Afghanistan still suffers with the poorest health indicators, such as the highest maternal mortality ratio of 327 deaths per 100,000 live births in the regions.

Although substantial numbers of people are known to be exposed to the dual consequences of complex emergencies and their additional impacts on the fragile health system, little is known about context-specific factors influencing access to healthcare in the crisis-affected populations. Moreover, we do not know about how these populations are more marginalized under Afghanistan’s health system, and whether and how innovative humanitarian programs, such as cash transfer programs, can fill their healthcare gaps. Using the dataset from ‘Whole of Afghanistan Assessment (WOA Assessment) 2019’, this study aimed to examine demographic, socioeconomic, and implementation barriers to health care in the crisis-affected population and to identify context specific modifiable factors for future humanitarian programs and health system reforms in Afghanistan.

## 2. Methods

### 2.1. Design and Data Collection

‘Whole of Afghanistan Assessment (WoA Assessement) 2019’ was designed as a descriptive study to investigate the humanitarian needs of the population affected by the crisis. This is a multi-sectoral needs assessment was carried out within the framework of the Inter-Cluster Coordination Team (ICCT) and co-facilitated by Renewed Efforts Against Child Hunger and undernutrition (REACH) [18], in close collaboration with the United Nations Office for the Coordination of Humanitarian Affairs (OCHA). Between 17 July and 19 September 2019, structured household surveys were conducted using Open Data Kit (KoBo Toolbox) amongst a sample of 31,343 displaced and crisis-affected households in accessible areas throughout all 34 provinces in Afghanistan. The household survey was based on a random cluster sample.

### 2.2. Variables

The variables included the demographic characteristics of the household affected by the crisis (number of family members, age, marital status) and the vulnerability of the households’ head (female household head, sensory difficulty, movement difficulty, cognitive difficulty, chronic illness). Displacement status of the household was also asked. Socioeconomic factors included highest education level achieved by household members, literacy (at least one household member over the age of 10 can read and write), employment status (number of adults worked outside), total monthly income, total debt, bank account (at least one household member has a bank account registered to their name), and phone with a subscriber identification module (SIM) card in household. The mean income by main source for families of Afghanistan was collected by their working and most of the spending of each Afghan family was categorized by items necessary for living. We also assessed whether respondents knew of information on assistance or were aware of humanitarian assistance availability, as indicator of the Afghan family support system. To assess access to health care facilities, respondents were asked: In the past 3 months, have you had access to a comprehensive health center in your village or close to your village where you could receive healthcare services?”, with response choices of “yes” or “no”; furthermore, the distance to healthcare facilities was also collected.

### 2.3. Ethical Consideration

Ethics approval for this human-subject database research was obtained from the institutional review board of (IRB No. DKU-IRB-NON2020-006) Dankook University.

### 2.4. Statistical Analysis

Descriptive analyses were conducted to summarize participants’ general, socio-demographic, and health-related characteristics related to access to healthcare facilities. Results are presented as sample frequencies with weighted percentage or weighted mean estimates with 95% confidence intervals as appropriate. Inter-group comparisons were performed using the Rao–Scott corrected chi-square test for categorical data, and sampling design weighted univariable linear regression for continuous data [19]. We performed multivariable binary logistic regression analysis to explore the factors related to healthcare facility access. The adjusted odds ratio (OR) with 95% confidence intervals (CI) estimates are reported. To produce unbiased estimates of population characteristics, the Taylor-series linearization was employed for variance estimation to account for the complex sample design of WoAA 2019 survey [20]. The data were analyzed using Stata/MP version 16.1 (StataCorp LP, College Station, TX, USA). The alpha level of 0.05 (two-tailed) was considered as a threshold for statistical significance.

## 3. Results

A total of 31,343 representative samples of households affected by the crisis in Afghanistan were included in this analysis. Among those crisis-affected households, 61.3% were of displaced populations originating from regions other than those where the interviews were conducted. The mean age of the household head was 43 years old, and 91.0% were ever married. Most households were headed by a male, while only 8.9% were headed by a female. The proportion of males to females was 48% females and 52% males in all age groups. The mean number of family members was 6.2. Education level was low, as 42.6% responded that they had never been educated. Only 56.0% of households had at least one individual over the age of 10 who could read and write. Regarding economic status of households, 15.8% did not have adults working outside, and the mean total monthly income was AFN 10,800 (USD 138.6) [21]. Total average debt of households was AFN 66,000 (USD 847.2) and health expenditure was reported as the most common reason. Approximately 5.1% of households had at least one person who had a bank account in their own name. For communication and access to information, 53.1% owned a phone with a SIM card registered to at least one family member. About 10% of the households reported having information about assistance or being aware of the current availability of humanitarian assistance. Related to disability of the household head, respondents reported difficulties in sensory function (seeing or hearing with aid, 15.8%), motor function (walking, 15.6%), or cognitive function (memory, concentration, or self-care, 19.9%). Despite the low rate of screening and diagnosis of non-communicable disease in Afghanistan, 19.8% responded that they had a chronic illness. For the distance to health care center, 77.8% of households were within a 5 km radius of healthcare facilities.

Of 31,343 households exposed to complex emergencies in Afghanistan, 10,057 (32.1%) could not access a healthcare facility when it was needed in last three months. With regard to accessibility to health services, it was identified that these demographic (number of family members, marital status) and socioeconomic factors (education, literacy, job status, total monthly income), communication and access to information (phone with a SIM card, knowing about information on assistance, awareness of humanitarian assistance availability), the distance to a healthcare facility, and displacement status resulted in significant differences (*p* < 0.05) (Table 1).

In order to explore the relating (predicting) factor for health care facility access in the crisis-affected households, multivariable binary logistic regression was performed (Table 2). Among households exposed to any type of complex emergency, their displacement status was significantly associated with access to healthcare (non-displaced vs. displaced, OR = 1.369, *p* = 0.008, 95% CI = 1.086 to 1.726). Economic factors such as employment status (one adult: OR = 1.448, *p* = 0.001, 95% CI = 1.158 to 1.811; two or more adults: OR = 1.605, *p* = 0.033, 95% CI = 1.039 to 2.480), and total monthly income (OR = 1.016, *p* < 0.001, 95% CI = 1.008 to 1.025) were associated with access to healthcare when it was needed. Significant increases were found in households with positive factors related to communication and access to information, such as awareness of humanitarian assistance availability (OR = 1.642, *p* < 0.001, 95% CI = 1.234 to 2.186) and phone with a SIM card (OR = 1.692, *p* < 0.001, 95% CI = 1.299 to 2.204). Meanwhile, disability of household head in cognitive functions such as memory or concentration (OR = 0.606, *p* = 0.039, 95% CI = 0.377 to 0.974) were associated with lower access to healthcare. Not surprisingly, compared with those living within 5 km of healthcare facilities, a greater distance was associated with a significant decrease in access to care (6 to 10 km: OR = 0.463, *p* < 0.001, 95% CI = 0.339 to 0.631; >10 km: OR = 0.374, *p* < 0.001, 95% CI = 0.277 to 0.504).

## 4. Discussion

Over the last decade, relative stability and accelerated improvements have been made in Afghanistan’s health system. Since 2003, the Basic Package of Healthcare Services (BPHS) and Essential Package of Hospital Services (EPHS) have been implemented to provide essential healthcare service to the entire population. The number of functional health facilities has increased from 496 in 2002 to 3135 in 2018, providing nearly 87% of the population with access to health facilities that are within two hours distance by any means of transport [22]. Health system improvements have substantially resulted in improvements in key health indicators, such as under-five mortality reduced from 191 per 1000 live births in 2006 to 50 per 100 live births in 2018 [23]. The progress in healthcare provision is undoubted, despite significant gaps that still remain.

These improvements in the health sector, however, have not been evenly distributed across social, economic, and regional strata. Health disparities persist or are even aggravated among marginalized groups in Afghanistan. Distance to health facilities, high medical cost, insecurity, rural and urban disparities, and gender inequality have been reported as barriers to access to health service [8,22]. In particular, as this study focused, the crisis-affected population is at serious risk of exclusion from the health system progress that has been accomplished in the last decade [15]. Access to essential public health was limited in more than thirty percent of the population in hard-to-reach regions living under insecurity [23].

Our analysis of Whole of Afghanistan Assessment (WoA assessment) 2019 confirms that there was substantial inaccessibility to healthcare in the crisis-affected population. After decades of conflict, one-third of crisis-affected households could not access healthcare facilities when needed. In the crisis-affected population, significant disparities were found among those who experienced displacement. Although 77.8% of the respondents live within 5 km of healthcare facilities, a greater distance to a health facility was significantly related to poor access to healthcare, as it related to more expenditure for transportation, which would constitute a significant barrier given the extreme poverty in the region [6,7]. Compared with disability rates in general population, such as functions of seeing (4.8%), hearing (2.0%), walking (11.9%), and remembering (5.3%) [22], 15.8% of household heads in the crisis-affected population reported disabilities in sensory function and 15.6% reported disabilities in motor function. Additionally, 19.9% suffered from impaired cognitive function. All disabilities resulted in increased vulnerabilities of households in terms of livelihood, employment, health, and nutrition.

Although overall health financing has risen substantially to USD 169 million in 2012 [24], Afghanistan’s health system has relied heavily on international donors, such as USAID, EU, and World Bank, accounting for 19.4% of total health expenditure, leaving only 5.6% from the government-financed public sector [25]. The rapid expansion of the for-profit health sector has resulted in parallel informal payments and out-of- pocket expenditure to fill the financial gap, with 75.5% of total health expenditure being spent by households [25]. The expansion of health markets is believed to have improved the health service coverage in the hard-to-reach areas, but has created financial barriers in the provision of essential health care [26]. This study shows that the poor economic status of the crisis-affected population, indicated by lower monthly income, is significantly associated with inaccessibility to healthcare in the crisis-affected population. Households with more employed adults have significantly better access to care.

These findings of economic barriers imply the need of and strategies for a multisectoral approach, especially cash transfer programs, in consideration of the aggravating poverty and livelihood struggles in Afghanistan [13]. Between 2007 and 2016, people living below the poverty line has more than doubled, affecting an estimated 80 percent of the Afghan population [9]. In 2019, entrenched conflicts and violence, exacerbated by natural disasters of flash flooding and chronic droughts, left the most vulnerable groups to face complex burdens of displacement, unemployment, and chronic illness and disabilities, all of which exacerbate poverty. In Afghanistan’s health system, which relies on out-of-pocket expenditure, coverage of essential health service is difficult to increase among populations living in growing poverty. As the study results show, additional costs, such as transportation expenditure for health facilities, would be significant economic barriers. Given the multisectoral impact of extreme poverty in healthcare and other social determinants of health [10], it would be unavoidable for humanitarian actors to operate comprehensive financing models such as conditional or unconditional cash transfer programs. However, the design and implementation of cash transfer programs need careful consideration given the heterogeneous and still uncertain impact on diverse operational contexts in low- and middle-income countries [12,27,28]. In particular, context-specific financing models are required in the conflict-affected regions, where pursuit of a livelihood is often limited by security concerns.

In Afghanistan, the operational health system was not established by the government’s own capacity but through extensive partnerships with NGOs. A provision of essential healthcare, such as the Basic Package of Healthcare Services (BPHS) and Essential Package of Hospital Services (EPHS), was outsourced to 40 national and international NGOs in 31 out of the 34 provinces. As this study indicates, factors related to access to information of assistance, especially awareness of humanitarian assistance availability potentially related to those non-governmental providers, are significantly associated with access to healthcare. Meanwhile, the disability in cognitive functions such as memory or concentration, potentially related to poor access to relevant information or communication, are negatively associated with healthcare access. Significantly, this study’s results highlight that mobile phones were associated with significant improvements in access to care in Afghanistan’s health system. In the fragile transition of a health system, healthcare barriers are not only rooted in limited resource but also in the instability and discontinuity of care provided by separated healthcare providers in public and private sectors. Mobile phones have great potential as essential communication tools for connecting patients to decentralized healthcare resources more effectively and delivering relevant health information in more affordable ways [29,30,31]. Recent studies also support the potential impact of mobile phones in the growth of household economies, gender equality, and empowerment [32]. To mitigate negative consequence of complex emergencies on depleted economic resources and healthcare, our findings suggest strengthening communication and access to information on assistance, especially through mobile phones. Particularly given the aggravating human immobility under COVID-19 pandemic, an innovative health financing system using mobile communication and cash transfer programs would be considered critical for filling the healthcare gaps in the conflict-affected but digitally connected population in Afghanistan.

## 5. Conclusions

In conclusion, despite undoubtable progress in recent years, Afghanistan’s health system still shares many features with post-conflict-affected zones in other parts of world, such as a high reliance of health financing from international donors, the dependency of health services on NGOs, and substantial out-of-pocket expenditures by households. Given the health system impact of high levels of insecurity and sharp increases in poverty, this study highlights that the crisis-affected population, especially displaced persons, are more vulnerable to exclusion from essential healthcare services provision. Financial barriers such as extreme poverty and unemployment are imposed more on the population affected by crisis and displacement, and thus impede their access to healthcare in the fragile health system. Therefore, digital determinants, such as access to a mobile network, along with addressing barriers to healthcare access, need to be reflected in the innovative financing models of future humanitarian programs.

## Figures and Tables

**Table 1 healthcare-09-00506-t001:** Characteristics Afghanistan crisis-affected households by healthcare facility access.

Variable	Category	Access to Healthcare Facilities				Total		*p*-Value
		No		Yes				
		*n*	Weighted%	*n*	Weighted%	*n*	Weighted%	
**Demographic and Socioeconomic Status**								
**Number of family members ***	(min = 1; max = 30)	5.9	(5.7, 6.1)	6.4	(6.0, 6.8)	6.2	(6.0, 6.5)	0.026
**Age ***	(min = 15; max = 110)	43.1	(42.6, 43.7)	43.8	(43.0, 44.7)	43.6	(42.9, 44.3)	0.070
**Marital status**	No	775	3.2	1511	5.8	2286	9.0	0.008
	Yes	8756	26.7	19,253	64.2	28,009	91.0	
**Gender of household head**	Male	9102	27.4	19,608	63.7	28,710	91.1	0.050
	Female	955	3.4	1432	5.5	2387	8.9	
**Education**	No	4890	16.3	6899	26.3	11,789	42.6	<0.001
	Yes	4687	13.7	13,980	43.8	18,667	57.5	
**Employment ****	0	2048	6.0	3284	9.8	5332	15.8	<0.001
	1	6473	21.1	14,253	49.6	20,726	70.6	
	≥2	1529	3.7	3498	9.9	5027	13.6	
**Total monthly income *^,†^**	(min = 0; max = 145.0)	8.8	(7.9, 9.7)	11.6	(10.1, 13.2)	10.8	(9.5, 12.0)	<0.001
**Total debt *^,‡^**	(min = 0; max = 49.9)	6.2	(5.3, 7.1)	6.8	(6.0, 7.6)	6.6	(6.1, 7.2)	0.432
**Bank account**	No	8848	28.5	19,121	66.5	27,969	95.0	0.648
	Yes	729	1.5	1758	3.6	2487	5.1	
**Displacement Status**								
**Displacement status**	Displaced	5001	20.7	9563	40.6	14,564	61.3	<0.001
	Non-displaced	4473	9.3	10,833	29.4	15,306	38.7	
**Communication and Access to Information**								
**Literacy**	No	4356	15.4	7438	28.7	11,794	44.0	<0.001
	Yes	4681	14.2	12,644	41.8	17,325	56.0	
**Mobile phone with a SIM card**	No	5555	17.6	8688	29.4	14,243	46.9	<0.001
	Yes	4022	12.4	12,191	40.7	16,213	53.1	
**Knowing of information of assistance**	No	8915	28.1	18,314	61.6	27,229	89.7	0.015
	Yes	1142	2.6	2726	7.7	3868	10.3	
**Awareness of humanitarian assistance availability**	No	9054	28.4	18,226	61.2	27,280	89.6	<0.001
	Yes	1003	2.3	2814	8.1	3817	10.4	
**Health Status *** and Healthcare Facilities**								
**Sensory dysfunction** **(Vision or hearing)**	No	8278	26.2	17,775	58.0	26,053	84.2	0.348
	Yes	1708	4.4	3209	11.4	4917	15.8	
**Motor dysfunction** **(Walking)**	No	8336	25.9	17,457	58.6	25,793	84.5	0.969
	Yes	1570	4.7	3403	10.8	4973	15.6	
**Cognitive dysfunction** **(Memory or concentration)**	No	7587	23.8	16,641	56.3	24,228	80.1	0.167
	Yes	2426	6.9	4385	13.0	6811	19.9	
**Chronic illness**	No	7546	24.1	16,664	56.1	24,210	80.2	0.357
	Yes	1876	5.6	3979	14.2	5855	19.8	
**Healthcare facility distance**	≤5 km	5908	21.2	16,124	56.6	22,032	77.8	<0.001
	6 to 10 km	2590	6.6	3494	9.5	6084	16.1	
	>10 km	1559	2.9	1422	3.2	2981	6.1	

* Weighted mean estimates and 95% confidence intervals are presented. ^†^ Per 1000 Afghani. ^‡^ Per 10,000 Afghani. ** Number of household members working in last 30 days. *** Health status of household head; SIM, subscriber identification module. Strata with single sampling unit centered at overall mean.

**Table 2 healthcare-09-00506-t002:** Factors influencing Afghanistan people’s access to healthcare facilities.

Variable	Category	OR	SE	*p*-Value	95% CI	
					LL	UL
**Demographic and Socioeconomic Status**						
**Number of family members**	(min = 1; max = 30)	1.015	0.049	0.753	0.924	1.116
**Age**	(min = 15; max = 110)	0.997	0.004	0.419	0.990	1.004
**Marital status**	No	ref				
	Yes	1.165	0.257	0.489	0.756	1.796
**Gender of household head**	Male	ref				
	Female	0.877	0.174	0.507	0.595	1.293
**Education**	No	ref				
	Yes	1.439	0.240	0.029	1.038	1.994
**Employment ***	0	ref				
	1	1.448	0.165	0.001	1.158	1.811
	≥2	1.605	0.356	0.033	1.039	2.480
**Total monthly income ^†^**	(min = 0; max = 145.0)	1.016	0.004	<0.001	1.008	1.025
**Total debt ^‡^**	(min = 0; max=49.9)	1.013	0.018	0.484	0.977	1.050
**Bank account**	No	ref				
	Yes	0.745	0.117	0.061	0.548	1.013
**Displacement Status**						
**Displacement status**	Displaced	ref				
	Non-displaced	1.369	0.162	0.008	1.086	1.726
**Communication and Access to Information**						
**Literacy**	No	ref				
	Yes	1.04	0.173	0.807	0.752	1.441
**Mobile phone with a SIM card**	No	ref				
	Yes	1.692	0.228	<0.001	1.299	2.204
**Knowing of information of assistance**	No	ref				
	Yes	0.790	0.123	0.130	0.581	1.072
**Awareness of humanitarian assistance availability**	No	ref				
	Yes	1.642	0.240	0.001	1.234	2.186
**Health status ** and Healthcare Facilities**						
**Sensory dysfunction** **(Vision or hearing)**	No	ref				
	Yes	1.656	0.597	0.16	0.817	3.359
**Motor dysfunction** **(Walking)**	No	ref				
	Yes	0.982	0.18	0.921	0.680	1.416
**Cognitive dysfunction** **(Memory, concentration)**	No	ref				
	Yes	0.606	0.147	0.039	0.377	0.974
**Chronic illness**	No	ref				
	Yes	0.912	0.086	0.328	0.759	1.097
**Healthcare facility distance**	≤5 km	ref				
	6 to 10 km	0.463	0.073	<0.001	0.339	0.631
	>10 km	0.374	0.057	<0.001	0.277	0.504

* Number of household members working in last 30 days. ** Health status of household head. ^†^ Per 1000 Afghani. ^‡^ Per 10,000 Afghani. OR, odds ratio; SE, standard error; CI, confidence interval; LL, lower limit; UL, upper limit; SIM, subscriber identification module. Strata with single sampling unit centered at overall mean.

## Data Availability

Data was obtained from REACH Initiative and are available with the permission of REACH Initiative.

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
