# Peer review of "Digital and Economic Determinants of Healthcare in the Crisis-Affected Population in Afghanistan: Access to Mobile Phone and Socioeconomic Barriers"

_healthcare, 2021, doi:10.3390/healthcare9050506_

Round 1
Reviewer 1 Report
The study is interesting and appreciable.
It is necessary stress and elucidate the implications.
Why this realty is significant for and can improve healthcare management within countries with weak healthcare systems and institutional problems?
Author Response
We do appreciate your positive comment. To emphasize the implications of our findings, we have revised the introduction and discussions with the revised title. In particular, as a key interpretation of our context specific findings, we have highlighted the digital determinants of healthcare such as access to mobile phone, along with traditional economic determinants in fragile health financing system related to income, employment, in order to address the humanitarian program with innovative financing model using mobile communication and cash transfer programs. Those digital and economic determinants that we have identified from large samples of humanitarian contexts would be significant evidence for the humanitarian program reforms. Please see more details in the revised manuscript we attached.
Reviewer 2 Report
“Access to mobile phone and socioeconomic barriers influencing access to health care in crisis affected population in Afghanistan” - A Referee Report:
----------------------------------------------------------------------------------------------------------------
The paper addresses a critical issue of improving access to healthcare in a war-torn country. This research is quite relevant, and its findings have merit, especially for the policymakers.
However, the sections on the introduction, conclusion and literature survey require work toward further improvement.
Major Issue:
----------------------
Authors should present their findings as further evidence supporting the idea popularized by two Nobel Laureates. The idea is that providing economic incentives, such as cash transfer, mobile clinics, cheaper communication tools, to people trapped in poverty facilitate their access to health care. Authors should incorporate that idea, the literature surrounding that idea and how their findings contribute to that literature. They should incorporate those changes to revise their abstract, introduction, and conclusion to enhance the value of this paper’s contribution significantly. In particular, they should review the following two critical articles from top core journals authored by two economists who recently won the Nobel prize for their contributions in similar areas:
- Esther Duflo. “Child Health and Household Resources in South Africa: Evidence from the Old Age Pension Program”, The American Economic Review, May, 2000, Vol. 90, No. 2, Papers and Proceedings of the One Hundred Twelfth Annual Meeting of the American Economic Association (May, 2000), pp. 393-398
- Abhijit V. Banerjee and Esther Duflo. “The Economic Lives of the Poor”, Journal of Economic Perspectives—Volume 21, Number 1—Winter 2007—Pages 141–167.
Authors should incorporate those two Nobel Laureates' findings to highlight some of the results reported in this paper but not discussed adequately.
For example, authors could highlight numbers from Table 2 to highlight the importance of cash transfer as an incentive to those trapped in poverty and mobile clinic to alleviate the transport problem in this regard.
Minor Comments:
---------------------
- Abbreviate the paper’s title to gain readership. How about “What works in promoting access to healthcare in a war-torn country like Afghanistan?” However, it is only a suggestion. The authors should feel no obligation to use it. The idea is to generate interests among potential readers.
- The abstract should be abbreviated. It should only answer three questions “what” is this research about, “why” should we care and “how” the results contribute to the literature. It does not require detailed statistics except the interpretations of those statistics.
- The introduction should include a research question such as “What works in promoting access to healthcare in a war-torn country like Afghanistan?” and motivate its importance by highlighting gaps in the literature. Avoid distractions. Make one point clear: tools for communication and travel cost and low income are the barriers; so, direct cash transfer and other economic incentives can help. Then reiterate the same point in the conclusion but this time point to Table 2 which shows that “healthcare facility distance”, “phone with a sim card” and “monthly income” are highly significant. Then offer that as evidence for motivating a policy of “cash transfer” or “mobile clinic” to alleviate those barriers.
Author Response
- Major Issue: Authors should present their findings as further evidence supporting the idea popularized by two Nobel Laureates. The idea is that providing economic incentives, such as cash transfer, mobile clinics, cheaper communication tools, to people trapped in poverty facilitate their access to health care. Authors should incorporate that idea, the literature surrounding that idea and how their findings contribute to that literature. They should incorporate those changes to revise their abstract, introduction, and conclusion to enhance the value of this paper's contribution significantly. In particular, they should review the following two critical articles from top core journals authored by two economists who recently won the Nobel prize for their contributions in similar areas:
- Esther Duflo. "Child Health and Household Resources in South Africa: Evidence from the Old Age Pension Program", The American Economic Review, May, 2000, Vol. 90, No. 2, Papers and Proceedings of the One Hundred Twelfth Annual Meeting of the American Economic Association (May, 2000), pp. 393-398
- Abhijit V. Banerjee and Esther Duflo. "The Economic Lives of the Poor", Journal of Economic Perspectives-Volume 21, Number 1-Winter 2007-Pages 141-167.
Authors should incorporate those two Nobel Laureates' findings to highlight some of the results reported in this paper but not discussed adequately. For example, authors could highlight numbers from Table 2 to highlight the importance of cash transfer as an incentive to those trapped in poverty and mobile clinic to alleviate the transport problem in this regard.
-> We appreciate for your guidance on the significance of economic barriers and incentives for access to healthcare in the marginalized population trapped in poverty. The two papers of Abhijit V. Banerjee and Esther Duflo, especially “the Economic Lives of the Poor” was very helpful to understand the vulnerabilities and some resilience of those who were in extreme poverty. We fully agree with your emphasis on cash transfer in related to economic barriers in fragile health financing and extreme poverty in Afghanistan. Based on our better understanding from these two papers, we have reviewed the cash transfer programs in Afghanistan and other operational contexts and revised the introduction and discussion (and title and abstract) to reflect the implications of our study findings more on cash transfer as innovative solutions. The revised manuscript cited two articles reviewer suggested along with several articles and report about cash transfers. Please see more details in the attached manuscript.
- Minor Comment 1: Abbreviate the paper's title to gain readership. How about "What works in promoting access to healthcare in a war-torn country like Afghanistan?" However, it is only a suggestion. The authors should feel no obligation to use it. The idea is to generate interests among potential readers.
-> We have revised the title to generate more interests among potential readers, as a “Digital and economic determinants of healthcare in the crisis affected population in Afghanistan: Access to mobile phone and socioeconomic barriers”. Following your suggestion, we tried to gain more readership by emphasizing our two key findings of digital (access to mobile phone) and economic (employment, monthly incomes etc.) determinants in a war-torn country like Afghanistan. These two determinants were highlighted for addressing needs for cash transfer for those economically marginalized but digitally connected population.
- Minor Comment 2: The abstract should be abbreviated. It should only answer three questions "what" is this research about, "why" should we care and "how" the results contribute to the literature. It does not require detailed statistics except the interpretations of those statistics.
-> Abstract has been abbreviated with less details of statistics. We have also attempted to clarify the answers with additional sentences, although word limits in the abstract was challenging.
- Minor Comment 3: The introduction should include a research question such as "What works in promoting access to healthcare in a war-torn country like Afghanistan?" and motivate its importance by highlighting gaps in the literature. Avoid distractions. Make one point clear: tools for communication and travel cost and low income are the barriers; so, direct cash transfer and other economic incentives can help. Then reiterate the same point in the conclusion but this time point to Table 2 which shows that "healthcare facility distance", "phone with a sim card" and "monthly income" are highly significant. Then offer that as evidence for motivating a policy of "cash transfer" or "mobile clinic" to alleviate those barriers.
-> Thanks to your comment, we have reviewed the cash transfer programs in Afghanistan and other humanitarian contexts and found its strength and limitation of available evidence especially in the context of Afghanistan. In order to emphasize the needs for cash transfer from our findings of economic and digital determinants of healthcare, we have tried to revise the introduction and discussions for example as follows (but not limited to). Please see more details in the revise manuscript:
- (Introduction) “In spite of substantial progress in the war-torn health system, Afghanistan still faces health challenges due to lack of sustainable health financing and workforce and the high reliance of international donors [8]. Last decade, economic hardships has been aggravated in the households impacting on social determinants of health through complex pathways in those trapped in poverty. [21] The epidemiological transition with non-communicable disease has been additional burdens on the rapidly growing population. In a fragile health system without sustainable health financing, health services are often delayed or inaccessible, resulting in preventive deaths or chronic diseases that have lost income and higher healthcare costs [6]. Given those economic barriers and extreme poverty, cash transfer programs have been considered as innovative intervention models to increase human capital in poor household. In Afghanistan, international actors have operated pilot cash transfer programs although they are still limited with risks of corruption, diversion, inflationary effects in complex operational environments.”
- (Discussion) “These findings of economic barriers imply the needs of and strategies for cash transfer programs, especially considering the aggravating poverty and livelihood in Afghanistan. Between 2007 and 2016, people living below the poverty line has been more than doubled estimating 80 per cent of the Afghan population [21]. In 2019, entrenched conflicts and violence, exacerbated by natural disasters of flash flooding and chronic droughts, left the most vulnerable groups to complex burdens of displacement, unemployment, and chronic illness and disabilities that exacerbate poverty. In Afghanistan health system relying on out-of-pocket expenditure, coverage of essential health service is hard to be increased among populations in the growing segments of poverty. As the study results showed, additional costs such as transportation to health facilities would be significant economic barriers. Given the multisectoral impacts of extreme poverty in healthcare and other social determinants of health, it would be unavoidable for humanitarian actors to operate comprehensive financing models such as conditional or unconditional cash transfer program. However, the design and implementation of cash transfer programs need careful consideration given its heterogeneous and still uncertain impacts in diverse operational contexts in low- and middle-income countries. In particular, context-specific financing models are required in the conflict-affected regions where livelihood is often limited under security concerns.”